# Two-Step Approach to Processing Raw Strain Monitoring Data for Damage Detection of Structures under Operational Conditions

**DOI:** 10.3390/s21206887

**Published:** 2021-10-17

**Authors:** Peng Ren, Zhi Zhou

**Affiliations:** 1School of Civil Engineering, Dalian University of Technology, Dalian 116081, China; 2School of Civil Engineering, Hainan University, Haikou 570228, China; zhouzhi@dlut.edu.cn

**Keywords:** structural health monitoring, damage detection, strain sensor, data interpretation, masking effect, decoupling, low-rank property

## Abstract

Strain data of structural health monitoring is a prospective to be made full use of, because it reflects the stress peak and fatigue, especially sensitive to local stress redistribution, which is the probably damage in the vicinity of the sensor. For decoupling structural damage and masking effects caused by operational conditions to eliminate the adverse impacts on strain-based damage detection, small time-scale structural events, i.e., the short-term dynamic strain responses, are analyzed in this paper by employing unsupervised modeling. A two-step approach to successively processing the raw strain monitoring data in the sliding time window is presented, consisting of the wavelet-based initial feature extraction step and the decoupling step to draw damage indicators. The principal component analysis and a low-rank property-based subspace projection method are adopted as two alternative decoupling methodologies. The approach’s feasibility and robustness are substantiated by analyzing the strain monitoring data from a customized truss experiment to successfully remove the masking effects of operating loads and identify local damages even concerning accommodating situations of missing data and limited measuring points. This work also sheds light on the merit of a low-rank property to separate structural damages from masking effects by comparing the performances of the two optional decoupling methods of the distinct rationales.

## 1. Introduction

Structural health monitoring (SHM) that consists of multidisciplinary technologies such as sensor, data processing, computer modeling, and mechanics inverse analysis can be responsible for the aging of infrastructures with the advantage of reducing the cost of the visual-based inspection and raising the efficiency of safety assessments. For the past two decades, there has been a rapid rise in the use of the civil SHM system [1,2]. The popularization and use of intelligent sensors have been increasingly common on bridges, tunnels, dams, tall buildings, and long-span spatial structures, thus gaining a large amount of operational data of structural facilities. Making full use of SHM data to serve structural health diagnosis and prognosis effectively has become a recent trend in this field [3,4,5,6].

In fact, SHM data interpretation to reveal the structure’s health state has received much attention over the last two decades, though most studies only used limited data. These data interpretation strategies could be roughly divided into two families: model-based and data-driven strategies [7,8]. The former generally depends on the accurate finite element model of a real structure and limited vibration observation to obtain its numerical updated version to quantify damage but often requires additional computation to solve the inverse problem. On the other hand, the latter aims to extract the damage-sensitive features and classifications to facilitate achieving SHM’s objective of online diagnosis [9]. However, previous SHM activities have laid particular stress on technologies of interest, such as testing novel sensing systems, while many failed to follow up corresponding diagnoses and prognoses and face up to the challenges of data interpretation in situ. Researchers from References [4,10,11] boiled down the challenges in this field to two aspects: incomplete and erroneous monitoring data and the coupling effects of local damage and time-varying (e.g., thermal, traffic, etc.) loads. Lately, machine learning tools are anticipated to help understand structural behaviors and select key attributes from massive operational data to improve data-driven strategies [4,5,12], such as using the support vector machine [13], principal component analysis [14,15], low-rank and sparse optimization [16,17], deep learning [18], and computer vision [19,20] methods. These methods that can often be used for big data analysis can also offer potential solutions for the data management and maintenance of in-service infrastructures. In this context, this work focuses on fixing the second challenge mentioned above to a certain extent.

The inevitable damage accumulation in the in-service infrastructure firstly gives rise to the local performance degradation of the structural components/system. Since strain monitoring data from SHM activities can directly relate to the stress redistribution in the vicinity of the strain sensor, such as damage initiation and development, strain-based damage detection approaches are recognized as great importance to structural health diagnosis [21,22,23,24]. Structural strain monitoring data could be divided into quasi-static and dynamic (high-frequency) ingredients, producing corresponding damage detection approaches [25,26,27,28]. As far as the theoretical approaches are concerned, the significant change of strain data or strain-based feature data will appear when there is damage triggering. However, due to the homologous characteristics of structural responses that reflect changing environmental and operational conditions, the strain-based damage detection approaches also face the above second challenge of SHM data interpretation [29,30]. Therefore, time-varying impacts on strain responses should be accounted for to avoid masking the effects caused by structural damage.

Conventionally, supervised models can be constructed by the regression between environmental parameters and structural damage-sensitive features to reduce the adverse impacts of these time-varying conditions in a structural health diagnosis. Much research considers the corresponding physical principle as a black-box model and assumes that all its parameters can be determinate from the training data, such as utilizing the measured temperatures and quasi-static strains in a long-term period [30,31]. Nevertheless, the influence of environmental factors (e.g., temperature and humidity) on the observed damage-sensitive feature data may be physically very complex and often not fully understood [32,33,34]. For example, the thermal-related strain of each rod in the truss bridge may be dominated by the specific temperature gradient [35], which makes it challenging to construct a unified explicit model. In addition, there are not sufficient sensors to meet the complete long-term observations of environmental conditions of infrastructure, such as its temperature distribution and radiation parameters, thereby increasing the model uncertainty in the structural health diagnosis.

An alternative data-driven strategy uses environmental effects as latent variables by employing unsupervised multivariate statistical tools. Following the orthogonal projection methodology, such as the principal component analysis (PCA), the new multivariate data in the hyper-plane projection space could be split into two distinct ingredients to represent the effects of dominant environmental factors and noises or anomalies, respectively [36,37]. Yan et al. utilized PCA to define the vibration features identified at different instants of the monitoring data under the linear or weakly nonlinear cases to distinguish between changes due to environmental variation and structural damage by the novel damage indicator (DI) [38]. Posenato et al. proposed to apply a sliding time window to quasi-static strain data set, resulting in the Moving PCA method [39], which executes PCA using only the latest window-sized data to obtain DI from the principal component directions and, thus, allows the online damage detection. However, since the latent variables assumption exists in the multivariate statistical tools, the underdetermination and overestimation of the damage are still inevitable due to the physical complexity and the limited measuring points. For this reason, Zhu et al. proposed to use independent component analysis to screen the combinations of strain sensors to get the ones with the highest correlation between temperatures and quasi-static strains before the Moving PCA was executed for anomaly detection [40]. The strategy enhances the sensitivity of anomaly detection and eliminates the delay from the application of the Moving PCA only. Liu et al. combined strain sensors in different bridges into various clusters in a similar thermal-related strain probability distribution [41]. The subsequent damage detection step was carried out on each independent cluster, in which the DI based on the probability distribution of strain monitoring data is insignificantly affected by the environmental temperatures. In general, although decoupling of damage and thermal effects is currently considered concerning strain monitoring data, the challenges related to complex physical principles and dependence on environmental information still exist.

In this paper, an idea that offers data interpretation in a distinct time scale is employed. Short-term dynamic strain response data dominantly rendered by the operating loads are applied to the analysis. Due to the fact that the frequency of environmental conditions (e.g., thermal effects) is considerably lower than the frequency of structural vibration under operating conditions (e.g., traffic), the analyzed signals for a shorter temporal length no longer depend on the temperature-dominated correlation [42,43,44]. Meanwhile, the high-frequency ingredient in the raw strain monitoring data can be apt to be separated from the ingredient of the daily temperature variations by the wavelet or other time–frequency transform tools [29]. Moreover, when some operating loads dominate the target structure at a series of small time-scale periods, the structural responses are likely to behave regularly, facilitating damage-sensitive feature extraction and the subsequent decoupling. Hence, environmental information such as temperature may not participate in data interpretation in such time sequences.

In this context, there is a two-step approach brought forward in this work to processing raw strain monitoring data only under operating loads for damage detection: in the first step, since relatively high-frequency dynamic strains increase the amount of data, the wavelet analysis tool is first used to process the strain responses to achieve initial feature extraction; in the next step, two data-driven methodologies, PCA and another one denominated as the low-rank subspace projection are presented and applied in this work, respectively, for decoupling the effects on the strain-based feature data of the operating loads and anomalies (probably structural damages), thereby getting corresponding local and global DI values. On these bases, an output-only damage detection strategy is proposed, consisting of data collection, anomaly detection and the above two key steps and executes these procedures efficiently and successively in the sliding time window.

To validate the damage detection strategy developed, we customize a steel truss model and its reaction frame system for continuous random excitations that can simulate time-varying operating loads, thus acquiring several independent vibration data to form the raw strain monitoring data set analyzed. The proposed two-step approach is carried out to obtain the global DI values and their outlier analysis results in each sliding time window and to exhibit the local DI values once the damage alert is raised. The results show that the strategy can detect the damage presence in the truss structure and the evolution from one to another damage state in time. The effects of damage localization from the two alternative decoupling methods are fully compared in the experiment. Moreover, the performance of the damage detection strategy is further evaluated concerning the cases of missing data and limited sensor deployments.

## 2. Approach

Figure 1 shows the output-only damage detection strategy consisting of two key steps: initial feature extraction and decoupling (or named local and global DI calculation), as well as the other two essential steps/modules: data acquisition and outlier detection, which involve how to acquire data with appropriate intervals and perform outlier analysis to output results of damage detection, respectively. Note that a sliding time window is applied to the strain monitoring data in this study, which has been proven to enable the procedure to iteratively calculate the above main modules, with less computational cost and more timely capture of structural damage presence [39,40]. A detailed description of the damage detection strategy is given in the subsequent implementation section.

The two-step approach for data interpretation refers to Steps 1 and 2 shown in Figure 1. Step 1 performs a feature extraction procedure by the wavelet tool, allowing dynamic strain responses to be decomposed into frequency band components on the specific scales. The initial feature can be extracted in this step from an appropriate frequency band, thereby assisting data reduction and noise reduction. Step 2 adopts data-driven, i.e., unsupervised multivariate statistical tools to deal with the initial features obtained from Step 1 for decoupling the effects of operating loads and structural damage. As a result, the local and global DI values corresponding to data-driven methodologies are calculated, sensitive to local damages but insensitive to operating loads. The rationales of the methods for Step 1 and Step 2 are described next.

### 2.1. Wavelet-Based Signal Feature

The wavelet theory provides a time–frequency analysis tool for both stationary and nonstationary signals. One of the main goals of wavelet analysis is to extract useful information from raw data, such as signal features and image edges. Based on wavelet multiscale decomposition, one vibration signal can be characterized as many sequences with different levels and frequencies without losing any components. For example, bridge strain monitoring data can be conveniently decomposed into approximate and detailed components [29], separating quasi-static and dynamic strains. In this paper, the wavelet packet tool quantifies the strain sequence into finer frequency components level by level for reconstruction. Our signal modeling proceeds very much in the same way as early theoretical documents, which were initially proposed by Mallat [45] and later developed by Meyer [46]. Therefore, we briefly describe the filtering results as follows:

A segment of strain sequence at the *i*th sensor channel, namely $\varepsilon_{i}$, can be equally divided into $2^{J}$ sub-frequency bands at the *j*th decomposition level in terms of the wavelet packet. $\varepsilon_{i}$ can thus be reconstructed as
(1)$$\varepsilon_{i}=\sum_{j=0}^{2^{J}-1}\varepsilon_{ij}=\varepsilon_{i0}+\varepsilon_{i1}+\cdots +\varepsilon_{i,2^{J}-1}$$

To provide a way of characterizing dynamics in the vicinity of sensor i, we chose one of the sub-band sequences in Equation (1) for analysis. The determination of the sub-frequency band j should follow two principles. First, as far as the low-pass filtering, i.e., denoising, is concerned, the sub-band sequence $\varepsilon_{ij}$ should be in the lower-frequency range. Furthermore, the selection of frequency band should be according to the maximum energy rule, because the maximum response amplitudes (energy) often lie on the frequency band where the dominant operating loads apply or the natural frequencies of the structure.

Consequently, at the *j*th decomposition level, we can extract the energy feature of the chosen sub-frequency band sequence $\varepsilon_{ij}$ in the raw strain monitoring data, namely wavelet packet energy strain or WPES for short and given by:(2)$$WPES_{ij}={\left\|\varepsilon_{ij}\right\|}_{2}$$
where ${\left\|\cdot \right\|}_{2}$ represents the Euclidean norm of the sub-band sequence. Equation (2) is also understood as the root mean square of the sub-band sequence $\varepsilon_{ij}$.

The WPES defined in this study involves a segment of strain sequence, and it is associated with both operating loads and structural intrinsic characteristic. Therefore, the feature extraction in this step only plays a role in data reduction and noise reduction but not damage alert, thus called the initial feature extraction (Step 1 in Figure 1). In addition, selecting an appropriate decomposition level is not so critical, as we may be analyzing strain monitoring data consisting of a large volume of data segments, including baseline data. Different decomposition levels J can be observed in these baseline data to obtain stable results when determining the sub-frequency band j.

### 2.2. Data-Driven Methodologies

#### 2.2.1. PCA

PCA methodology puts a way to reveal the relationship between the time-varying factors and the multivariate features extracted from SHM data [36,37,38,39]. The main operation of PCA is to maximize the variance of the orthogonal projection of the original data set if each dimension of the data set follows an independent Gaussian distribution. A brief description of the theory of PCA, for completeness, is presented in this section. Additionally, the reader can be asked to refer to References [47,48] on this topic.

In this decoupling step, we use a minor variation of the moving PCA’s procedure [39]. Specifically, in our procedure, we define the WPES as the initially extracted feature under operating loads and make the WPES data set, instead of quasi-static strain, lay a foundation for the DI calculations. For the sake of simplicity, let us use the generic X to represent the data set WPES.

If X is the data set with m variables (dimensions) and n observations (m≤n), the operation of the maximization of variance mentioned above could be deducted to be equivalent to solving the eigenvalues and eigenvectors of the covariance matrix of X. These eigenvectors are the basis vectors used for orthogonal projection, also known as the principal components or the principal component directions.

Meanwhile, the covariance matrix of X is equal to ${\hat{X}}^{T}\hat{X}$, where $\hat{X}$ is the sample mean of X with each of its columns centered. Computationally, the calculation of principal components can also be performed by a Singular Value Decomposition (SVD):(3)$$\hat{X}=U\Sigma V^{T}$$
where U and V are the unitary matrices comprising with left and right singular vectors, respectively, and obviously the right singular vectors are the above eigenvectors, i.e., principal component directions; Σ is a matrix of n rows and m columns but with m diagonal elements $\sigma_{i}$(i=1,⋯,m), in which $\sigma_{i}$ is called the singular value. Hence, the projected data on the principal component space is
(4)$$\hat{X}V=U\Sigma V^{T}V=U\Sigma$$

According to the cumulative contribution of variance, PCA takes the first d columns of $\hat{X}V$ to account for the dominant time-varying factors from Equation (4), i.e.,
(5)$$Y=\hat{X}[v_{1},\cdots ,v_{d}]=[u_{1}\sigma_{1},\cdots ,u_{d}\sigma_{1}]=U_{d}\Sigma_{d}$$
where Y is the projected data set or so-called principal scores in which the X is orthogonally projected on a *d*-dimensional hyper-plane, i.e., a new subspace. The selection of an appropriate dimension d is usually an issue needing attention. The left graph of Figure 2 instances a typical PCA dealing with two sets of sensor data considering both undamaged and damaged states. The variations of sensor data versus a dominant factor follows the positive linear correlation in the undamaged state, thereby in the sense of this case, d is equal to 1. Then, the damage initiation will change the previous relation so that it is possible to distinguish different states by the variations of the principal components.

However, when using PCA to analyze multivariate SHM data sets, the truncation order d will be more challenging to select. For example, the relationship between multivariate quasi-static strains and temperatures is likely to be complicated, and there may be multiple dominant factors. On the other hand, short-term dynamic strain responses are dominantly rendered by the operating loads such as traffic, train-bridge coupled excitation, or wind on large-scale flexible structures. As far as current studies [35,44] have also shown, the amplitude of the dynamic responses of a bridge structure under a single train load is virtually independent of temperature. Therefore, the dominant factor in the dynamic response at a small timescale is almost unique. In this sense, the value of the truncation order d is supposed to be 1.

As preceded by this section, PCA can then be used to process the multivariate WPES data set, and meanwhile, the latent variable reflects the embedded relation between the dominant factor in operating loads and the WPES features. In terms of Equation (5) and d=1, the first principal component, i.e., the first eigenvector $v_{1}$, represents the latent variable under operational conditions. Each element in the eigenvector corresponds to strain observations at a particular sensor location, i.e., a particular column of the WPES matrix. We thus define the element $v_{1}^{i}$ in $v_{1}$ as the local DI values, i.e.,
(6)$$DI_{local}=v_{1}^{i}$$

The corresponding global indicator is the Euclidean norm of $v_{1}$, i.e.,
(7)$$DI_{global}={\left\|v_{1}\right\|}_{2}$$

#### 2.2.2. LSP

This section presents another data-driven methodology that both the PCA and the low-rank property of the data structure [16,17] can use. Instead of using the multivariate statistical tool, our method is, to some extent, based on the low-rank property to treat pairs of initial feature data to construct the new feature. The advantage of our method is that the prejudgment of the truncation order d for the multivariate statistics, i.e., the number of the dominant factors in operating loads, is not required. More details on this are given below.

Figure 2 shows the two sets (or named a pair) of feature data in which the two-dimensional PCA is executed. Since the projection data $y_{ik}^{1}$ in the first principal component direction retains the dominant factor of the original data set $x_{ik}$, the first principal subspace where the projection is located is low-rank (the superscript of $y_{ik}^{1}$ represents the main subspace of the projection, and the rank is supposed to be 1 herein). It can be stated that the damage presence enables the data to not follow the low-rank subspace structure. The projection data $y_{ik}^{2}$ in the second principal subspace will change significantly from the undamaged to the damaged state, as shown in the right graph of Figure 2. The damage alert can be raised, because the change trend of $y_{ik}^{2}$ is not sensitive to the dominant factors but sensitive to structural damage in the vicinity of the *i*th or *k*th sensor.

Usually, several strain observations are obtained, and the WPES feature data can be extracted. Accordingly, we list the corresponding projection data in the second principal subspace, i.e., $y_{ik}^{2}$ in a matrix form at a particular sensor location i:(8)$$Y_{i}^{2}=[y_{i1}^{2},\cdots ,y_{i(m-1)}^{2}]$$
where $Y_{i}^{2}$ is the second principal subspace projection matrix corresponding to the *i*th strain sensor. The matrix $Y_{i}^{2}$ is calculated by the two sets of strain-based feature data in pairs rather than using multivariate statistical analysis. Consequently, we use the variances to characterize the variability of the projection data in the second principal subspace, i.e., representing the variation from the anterior low-rank property, hence denominating the vector forming by the column variances of the matrix $Y_{i}^{2}$ as the low-rank subspace projection (LSP) vector:(9)$$w_{i}^{2}=\left\{\mathrm{var}(y_{i1}^{2}),\cdots ,\mathrm{var}(y_{i(m-1)}^{2})\right\}$$

We define the expectation of $w_{i}^{2}$, i.e., $E(w_{i}^{2})$ as a local DI for a particular strain observation (sensor location), i.e.,
(10)$$DI_{local}=E(w_{i}^{2})$$

The corresponding global indicator is defined as the Euclidean norm of the vector consisting of $E(w_{1}^{2})$, i.e.,
(11)$$DI_{global}={\left\|\left\{E(w_{1}^{2}),\cdots ,E(w_{m}^{2})\right\}\right\|}_{2}$$

### 2.3. Implementation

As outlined in Figure 1, there are totally four functional steps/modules to execute the damage detection algorithm. Data acquisition module provides the basis for two-step data processing approach. A strain acquisition system can obtain structural dynamic strain responses under operational conditions with a sufficient dynamic sampling capability, e.g., using the packaged Fiber Bragg Grating (FBG) sensors with appropriate gauge lengths and their demodulation devices [26,28]. The sampling interval should be in the order of milliseconds, and the sampling frequency may be taken as around 200 or 500 Hz based on the actual structural vibration conditions and Nyquist sampling theory.

It is not easy to directly measure operational loads such as traffic, train-bridge-coupled excitation, or complex current fields. Therefore, only the dynamic components in strain monitoring data are imported in chronological order in the data acquisition module. The imported data should be short-term, i.e., second-order, but not necessarily continuous, and could be obtained through multi-resolution analysis of the raw strain monitoring data [29]. We employ sliding time windows, i.e., processing these data, at appropriate short windows and equal intervals to reduce computational cost and capture time of damage presence. The window size should be at least twice of the periodic variability [39]. The following formula denotes the window size $N_{w}$:(12)$$N_{w}=l\cdot n$$
where l is the same as the definition in Equation (2), which is the length of one strain sequence required for the initial feature (i.e., WPES) extraction. For example, for the bridge strain responses due to the train vibration load, l can be taken to be greater than the duration of each train passing, and n is the number of observations of train passing to form the WPES data set. In addition, the dimension of the WPES data set is denoted as m, which refers to the number of effective strain sensors deployed in the associated structure or substructure.

The subsequent two key steps are implemented in each sliding time window. The window is updated iteratively through each moving with l length. According to the above rationale sections, two optional data-driven methodologies are adopted and denominated as WPES-based PCA (or named PCA for short) and WPES-based LSP (or named LSP for short) methods, respectively. The proposed two-step approach is described as follows:Compute the WPES features according to Equations (1) and (2) with the Daubechies wavelet at appropriate scales. The Daubechies wavelet is often chosen due to its discrete wavelet counterparts used when embedding the algorithm at the sensor level.Obtain the global and local DI values for decoupling of feature data according to the procedures introduced in Section 2.2. The main calculation costs of the two optional methods include: performing SVD to get the first eigenvector for the PCA method and forming m−1 dimensional low-rank subspace projection vector based on Equations (8) and (9) for the LSP method.

For the purpose of damage detection, an outlier analysis is performed to classify between the undamaged and damaged states after the global DI values are obtained. Outlier analysis usually encompasses comparing global DI values to a set of control limits that are defined by the threshold based on the reference/baseline data obtained from the undamaged state. In general, the threshold is set to three times of standard deviation of the baseline data that should be determined in advance. Any future observations of global DI values outside the threshold (±3σ) can then be labeled as the one obtained from the damaged state. Meanwhile, once the damage alert is raised, the corresponding local DI values can be output the result—that is, the location of a strain sensor in the vicinity of which an abnormality occurs.

## 3. Experiment Design and Process

In laboratory, a reaction frame system shown in Figure 3a was customized to verify the proposed approach. A steel truss comprising 10 joints and 17 elements/rods was fabricated by approximately simulating a part of a truss bridge and fixed on the reaction system tightly with the four of the joints. The truss joints employ the pin–shaft connection and consist of roller bearings to maintain the free-rotating capability. Each joint has four roller-bearings in series for the mass consistent, as shown in Figure 3a in detail. The truss element/rod with a diameter of 5 mm has threaded connections with the roller bearings at its ends. We chose such ways of connections of joints on account of the fact that there would be only two force rods but no secondary stress in the truss structure. The length between the centers of two roller bearings is 200 mm of the horizontal and vertical elements, as well as 283 mm of all diagonal elements.

An air-actuated piston shaker was recruited for this vibration test, as shown in Figure 3a. The equipment allows to output axial motion in a particular force range, and although the force is uncontrollable, the output axial displacement is measurable by a linear variable differential transformer sensor and its signal amplifier and AD converter. Thus, by having the use of the air-actuated piston shaker, we were able to provide a series of vertical random excitations to simulate the short-term operating loads on the truss structure. For example, the excitations can be applied at node 5, as shown in the model sketch of Figure 4, plus four nodes with constrained ends to avoid the torsion effects caused by the out-plane motion.

During the experiments, the axial strains of all 15 stressed elements were measured by FBG sensors. The FBGs adhered to the rod’s middle and connected to the demodulator via the transmission fiber for synchronous data collection. The demodulator adopted was Model TFBGD-9000 (Harbin, China), as shown in Figure 3b, with a sampling rate of 200 Hz. Moreover, due to the adoption of the feature extraction step to reduce data of each sensor channel in the same excitation period, completely synchronized data collection is not necessary. Thus, in practice, the multivariate strain monitoring data can also be record asynchronously through multiple demodulation devices or channels.

Additionally, what we know about the difference between sensor temperature compensation and structural thermal effect is great in terms of the data processing in situ, although environmental temperature variations cause both. The former is because the temperature affects the central wavelength of the optical fiber sensor probe, which is an adverse impact that must be eliminated or can be easily eliminated. The latter is one of the problems to be solved urgently in SHM field, which leads to a perceptible quasi-static response of the structural component and should be accounted for to avoid masking the effects caused by structural damage. Since the time domain response of each excitation is short and the excitation starts from the zero equilibrium, there is no need to perform temperature compensation of the sensor. In addition, it was stated in the Introduction, existing studies have shown that the thermal effect has no significant impact on the vibration responses on a small timescale.

The damage was approximately simulated by a decrease in the cross-sectional dimensions of an element. As shown in Figure 4, Elements 6 and 15 were set as damaged elements that were replaced by a 4-mm rod. The first damage scenario (1st DS) starts once Element 6 was replaced with a lower stiffness rod at a certain excitation, and the second damage scenario (2nd DS) occurred when Element 15 was also replaced with a low stiffness rod at a subsequent excitation. That means, during the experiment, the truss structure underwent an evolution from an undamaged state to a single-damage state and then to the next damage state. Taking into account the performance of the exciter itself, we turned on each independent excitation for about 20 s and then cut them all into 11 s. A total of 30 independent excitations are selected to acquire dynamic strain response data. Among them, the first ten times correspond to the undamaged state of the truss structure, the medium ten times correspond to the first damage state and the last ten times correspond to the second damage state.

## 4. Results and Discussion

### 4.1. Initial Feature Extraction

Figure 5 corresponds to dynamic strain data set recorded in time and frequency domain from random vibration tests. A total of 330 s, i.e., 66,000 strain sampling points, were acquainted by the FBG sensors on each measured truss element at 30 independent excitations and measurements with the sampling rate of 200 Hz. Hence, considering all measured elements, a data set of the size 66,000×15 was obtained in the experiment for data interpretation.

As illustrated in Figure 5a, the strain responses of some elements only fluctuate in a small range at zero equilibrium. Those zero bars not stressed can be easily found according to the truss force analysis in theory. In practice, the zero bars could also be excited by only changing the force node in Figure 4. Nevertheless, we still used these zero-bar data as low noise inputs in the experimental data set to verify the robustness of the algorithm. In addition, it can be seen from Figure 5b that the strain data in frequency domain of all the elements are distributed around the bandwidth of 0–50 Hz, which is entirely the dynamic behavior dominated by the air-actuated piston shaker. These loads randomly generated by multiple independent excitations result in the strain responses of non-white noises, all the frequency bands of which we have used to demonstrate the SHM data interpretation under the operational condition.

Although the experimental data set has been displayed, we can implement the damage detection strategy introduced in Section 2.3 within the fixed size of the sliding time window instead of the whole time series. The parameter l from Equation (12) is set to 200, the same as the sampling frequency, making each extracted WPES value corresponding to exactly one second; the parameter n, i.e., the number of WPES obtained in each sliding time window is taken as 30, so the window size of the raw data is 6000 (i.e., 30 s). In the initial feature extraction step, the decomposition of the strain sequence is employed via the wavelet packet transform with five decomposition levels using the Daubechies 25 wavelet, thereby getting 32 sub-frequency bands.

The first 200 sampling points in the first sliding time window of all the strain sequences have been used as examples, as shown in Figure 6, where the strain sequence is named ss*i*, but for the sake of space, only ss1 and ss15 are shown. The maximum energy rule is applied for sub-band sequences to determine the extracted sub-frequency band and avoid noisy (high-frequency) ones. We have selected the fourth sub-frequency band, then used it to derive the initial feature WPES based on Equation (2).

Subsequently, the sliding time window is pushed forward by 200 sampling points, and the next WPES value is extracted, the corresponding WEPS data set is updated, and so on. Sampled data can be analyzed in real-time and output the detection results. A total of 330 initial features were extracted from each strain channel in the experiment, and each obtained WPES value corresponds to the time interval of one second. We can see from the WPES values of Figure 7 that the initial feature extraction step achieves data reduction and noise reduction compared to the raw data set.

Given that the introduced damage presences in the previous section, we have set the damage scenarios of the experimental truss with undamaged of 0–110 s, 1st DS of 111–220 s, and 2nd DS of 221–330 s. However, the WPES data set is typical of time-varying load-dominated response features, thus masking the effects caused by structural damage.

### 4.2. Detection Results

The decoupling step is executed instantly after the initial feature extraction step to calculate the DI and prepare the anomaly detection. A relatively small data set, which is the WPES matrix (30 × 15) herein, is available in each sliding time window, entailing less computational cost and more timely capture of the rapid change. There are two alternative ways to deal with this initial feature matrix. SVD is performed to get the first eigenvector and then to calculate the global DI based on Equation (7); the initial feature data are projected to their second principal subspace in pairs to form the LSP vector according to Equations (8) and (9) and, then, to calculate the global DI based on Equation (11). Figure 8 displays the temporal variation of the global DI values obtained by the two data-driven methodologies, respectively. The 80-s reference DI data from a stable state that has been confirmed in advance is utilized to establish the threshold, which is ±3σ over the reference data. Once the newly acquired DI exceeds the threshold (dotted lines in the subfigures), it is labeled as abnormal.

No matter which decoupling method is used, the global DI values obtained have experienced two large spikes synchronized with the damage initiations (shown as the colored vertical lines). These can also be observed in detail in the subfigure below. Both methods are able to use the ±3σ control limit to detect anomalies caused by damage since significant changes of the global DI values exceed the threshold after the damages are found. Compared with the above raw strain monitoring data and the wavelet-based strain response features only, our two-step approach clearly has an advantage over eliminating the effects of the time-varying loads and detecting damages. Moreover, observation shows that the spiking DI values will return to a stable state similar to the undamaged state after the damage initiation for just a time window size, which can be used to update the control limit for continuous monitoring and damage alert. Therefore, in this experiment, we can use the indicators presented to detect the damages of Elements 6 and 15 successively.

In addition, we found in Figure 8 that the change trends of the global DI values obtained from the two data-driven methods are reasonably different. It is because of the distant rationales of the two. PCA usually employs the first principal component direction $v_{1}$ as the latent variable to associate the operating load. The damage presence may bring about changes in the embedded relation. Thus, the trend of PCA-based DI values will accord with the relative change of this relation. On the other hand, LSP performs the two-dimensional PCA in pairs about the deconstruction of the original multivariate feature space and then adopts the projection data of their second principal subspace that reflect the low-rank property to draw the new DI values. The damage initiation undermines the anterior low-rank property, so that DI values increase significantly from approximate zero, and along with the stable state recovery, the DI values return to approximate zero again. The values of LSP-based DI values will consequently be absolute ones.

Figure 9 displays the temporal variation of the local DI values obtained by the two data-driven methodologies, based on Equations (6) and (10), respectively. The two dashed lines are included in the figure to separate the undamaged and two damaged scenarios in succession. From the plots, we can see that the local DI values of some elements, especially the damaged Elements 6 and 15, appear rapid changes after damage initiations, which concurs with the ability of the damage localization of the local DI values stated above. In line with the theoretical explanation about the two decoupling methods in the previous paragraph, the local DI values by PCA manifest the relative changes due to elemental damages, instead of the absolute ones by LSP with the fluctuations between zeros and spikes. However, both local DI values can suggest whether or not the structure reaches a new stable state.

For a clear visual comparison of the performance of local damage identification by initial WPES only, WPES-based PCA, and WPES-based LSP, we list the DI values of all elements at the moments of damage initiations (i.e., 1st DS and 2nd DS refer to 111 s and 221 s, respectively) in Figure 10 and Figure 11. It must be stressed that the local DI values by PCA manifest the relative changes due to damages, so identifying local damage needs the current value to be compared with a previous baseline. Otherwise, it is challenging to distinguish damaged elements only by using DI values at different elements, e.g., the DI values of different magnitudes when all elements at any fixed time are taken for analysis in Figure 9a. Therefore, this work uses the deviation obtained from the current DI value minus the baseline/reference value as the improved local DI by PCA in practice, and at the same time taking the consistency into account, the DI values by WPES-only also use the deviation ones. The reference time was first set to 31 s when gained the first local DI values.

It is evident from Figure 10a and Figure 11a that the WPES values, i.e., wavelet-based strain response features, cannot locate damages because the effects of operating loads are not eliminated. For the DI values by PCA, the results of damage identification rely on the selection of reference time, as shown in Figure 10b and Figure 11b,c. When the reference time is 31 s, the deviation of local DI value for Element 6 in Figure 10b is abrupt apparently and thus corroborates the 1st DS, though some elements, e.g., Element 8 or Element 14, are regarded as the positive faults. However, although the reference time is still 31 s, the deviations of local DI by PCA in Figure 11b do not corroborate the 2nd DS, which can be attributed to the baseline that is not updated in time after 1st DS and be classified as a negative fault in damage identification. Next, we updated the reference time to 181 s and then got the correct identification results with positive faults shown in Figure 11c. For the DI values by LSP, without the need for the baseline, the results in Figure 10c and Figure 11d clearly show that the damaged elements in the 1st DS and 2nd DS have been accurately identified with few positive faults. The LSP-based decoupling method permits the local DI of all elements to be directly put together for comparison.

Furthermore, the clarity is presented to characterize the performance of the proposed DI for conveniently and accurately identifying local damage (see Table 1). Note that the superscript 1 implies: high level means the damaged element is identified clearly by directly using the local DI with few positive faults; medium level means the improved local DI can identify the damaged element or with negative faults; low level means the damaged element is hard to identify or with many negative faults. The results show that although both decoupling methods are able to detect damage instantly, one marked observation to emerge from this experiment is that the better performance is achieved by means of the local DI based on the LSP rationale rather than PCA.

As a result, the outcomes of this experimental case study can be summarized as follows: First of all, the proposed output-only two-step approach processes the raw strain monitoring data efficiently from the vibration test to detect the elemental damages with the successful removal of the masking effects of operating loads, as shown in Figure 5, Figure 7 and Figure 8. Secondly, the comparison of the results from the adopted two data-driven methods in the decoupling step is of interest because of the finding of the different rationales that determine the performance of the two methods in local damage identification, as shown in Figure 9, Figure 10 and Figure 11. The DI by PCA is essentially associated with time-varying loads and relies on the baseline. Instead, the low-rank property of the DI by LSP makes it independent of the baseline and belonging to the well-done absolute measurement. The low-rank property of data structure may offer valuable insight into decoupling the structural damage and masking effects on SHM data.

However, although the proposed approach has drawn the global DI for damage alert and the local DI for identifying local damages, both of which can facilitate online diagnosis of structural condition employing unsupervised modeling, the performance of these indicators for indicating damage quantification has not yet been discussed in this experiment. The rods were replaced with weak uniform ones to simulate local damage in the experimental truss. Different degrees of damage are difficult to simulate in this study in practice. If targeting damage quantification, the numerical simulation based on a refined finite element model and appropriate supervised machine learning tools is necessary, which is too vital for further nuanced structural condition assessment.

### 4.3. Case for Different Sizes of Sliding Time Windows

A sliding time window is employed to calculate the global and local DI values successively in this paper. The raw strain monitoring data could be picked up in an equal interval according to Equation (12) when an appropriate time window is determined in advance. In the previous researches [39,40], the window size was selected based on experience. They pointed out that it should be greater than a multiple of the variation period and twice the fundamental period of the response is the shortest practical size. However, these researches targeted quasi-static strains related to long-term temperature variations but not like the vibration responses on a small time scale in this experiment. Therefore, different sizes are discussed in this section to show the impact of the size of the sliding time window on the damage detection results.

Four window sizes (10, 30, 50, and 70 s) used as an example are chosen at a specific interval and for a smaller length than the baseline data. The global DI values obtained by the WPES-based PCA and LSP methodologies are displayed in Figure 12. The damage initiations could be detected timely since significant changes of the global DI values can be observed when the 1st DS or 2nd DS occur. Although the change trends of the global DI values between the two data-driven methods are still different, the effects of the damage detection of both are not affected by the change of the window size. Next, for local DI values by PCA outlined in Figure 13a and Figure 14a, the reference times are uniformly chosen at 81 s for 1st DS and 151 s for 2nd DS. Concurred with our finding in the last section, we get negative faults in damage identification that cannot deal with the 2nd DS, especially for using the larger sizes (e.g., 50 and 70 s) of the sliding time window. On the contrary, for local DI values by LSP outlined in Figure 13b and Figure 14b, the results show that the 1st DS and 2nd DS can be accurately identified without the need for the baseline. From the above discussions, it would appear that it is legitimate to choose the window size of 30 s in this experiment considering smooth and accurate identification results.

Even though the raw strain monitoring data were utilized to substantiate the proposed damage detection strategy, the data set in this study might be restricted by a series of single loadings in the experiment. In addition, SHM with extensive data on the actual operation structures has very little public data-sharing, and especially the damage detection needs to reproduce the damage on operation structures to validate the algorithm, which is difficult to achieve at this stage. However, there are increasing cases of using strain sensors to monitor vulnerable parts of bridges [49,50,51] or weigh-in-motion [52,53,54] in actual engineering. An embodiment is the dynamic strain measurement when a train crossing a bridge, severely affected by the masking effect caused by typical operating loads. One promising application of our technique would thus be strain response data obtained from the train-bridge interaction system. At the same time, we believe that the sliding time window with an appropriate size may hope to tackle computational costs for the online diagnosis of structural conditions.

### 4.4. Case for Missing Data

In the SHM field, the issue of incomplete and erroneous monitoring data is widespread because of reasons such as inappropriate sensor deployment, sensor failure or replacement, poor quality of data transmission or power supply, and structural maintenance. Inside, the missing data issue may impede damage detection in time and lead to false analysis. There are currently two ways to deal with the missing data issue. One is the effective data cleansing or named augmentation algorithm that makes SHM data more reliable for online monitoring [20]. Another is that the algorithm itself that is robust can accommodate the situation of missing data [37]. This section primarily investigates the performance of the proposed two-step approach when suffering typical missing complete measurements, e.g., a period of structural maintenance that obstructed data transmission.

To simulate the case of continuous data missing, we chose a block of strain monitoring data and replaced it with null to represent missing values. Figure 15 shows the extracted WPES values from the rest of the sampling points. The treatment is that the 111–242 s, more than one-third of the raw data set, was compacted by removing the null data at all elements. In this example, the first part of measurements (1–110 s) is undamaged, and then after the measurement is restored, the structural state of the truss is unknown and needs to be detected at 243 s and afterward. We have known in the experiment that the two elements (Elements 6 and 15) of the structure are damaged, i.e., being in a double-damaged state at the detection time.

Figure 16 displays the results of damage detection, where the global DI values are calculated by the WPES-based PCA and LSP methods, respectively. Although the PCA methods in Figure 16a can find few delays, significant changes of the global DI values from both methods still exceed the ±3σ control limit (see the subgraphs) to detect the damages in time. Further damage localizations at 243 s show the double damages that occurred during data missing are identified (Figure 17). Note that random data missing or other sensor anomalies have not been considered yet in this study. However, the above results have further strengthened our conviction that the proposed two-step approach is robust to identify local damages.

### 4.5. Case for Limited Measuring Points

As the focus of the study was on the approach to processing the strain monitoring data, there is some likelihood that the strain-based DI could generally detect the local damages in the vicinity of the deployed strain sensors but not offer a distributed damage localization of a relatively large-scale structure. In this example, if the damage occurs outside the installed area, the proposed approach is also understandable to carry out structural damage detection using only a few measuring points rather than localizing the coordinate position of damage. We assumed that only a substructure is monitored (Figure 18) in the experimental steel truss, but the damage scenarios are consistent with the above cases. In addition, Element 11, a theoretical zero bar with only low noise outputs in the experiment, is also included in the monitored substructure, which adds to the difficulty of detecting damage.

As the damage detection results demonstrate in Figure 19, the proposed two-step approach successively detects the damages due to the stiffness losses of Element 6s and 15. Moreover, we have not discussed the optimal layout of strain sensors for damage localization in the future. The main difficulties include that the local damage is entirely random and cannot be predicted in advance. At the same time, the strain information only relates to the stress redistribution in the vicinity of the strain sensor. Despite this, Table 2 illustrates that the performance of the corresponding DI did not disappoint even concerning accommodating situations of limited measuring points. In our view, the results of the more stringent cases emphasize the feasibility and robustness of our two-step damage detection approach.

## 5. Conclusions

This paper presents a damage detection strategy based on data-driven modeling of the raw strain monitoring data. The key goal of the strategy is to develop a two-step approach to successively processing the short-term dynamic strain data in the fixed-size sliding time window and to decoupling structural damages and masking effects from operating loads in initial strain-based feature data (WPES). An experimental truss was customized with an emphasis on simulating time-varying operating loads through continuous random excitations. The proposed two-step approach’s feasibility and robustness were substantiated by analyzing experimental data under different damage scenarios.

The results have shown that both data-driven methodologies (PCA and LSP) in the decoupling step can perform well in detecting the damage presences through the corresponding global DI. Meanwhile, these DI values can return to a new stable state after damage alert to detect further damages. Nevertheless, in terms of DIs for local damage identification, it appears that the performance of damage localization based on the LSP rationale is better than that based on PCA. Unlike the PCA indicator, the low-rank property of the DI by LSP makes appear the fluctuations between approximately zeros and spikes when identifying local damages, thus making it independent of the baseline. Furthermore, the robustness of the proposed approach was preliminarily demonstrated in the experiment to accommodate situations when different sliding time windows were employed, measurements were missing entirely, and measuring points were limited.

From the SHM prospective, finding an indicator that is not sensitive to the effects of environmental and operational conditions and related only to the changes of structural parameters such as local stiffness is a critical part of achieving unsupervised structural condition assessment. Long-term environmental factors (e.g., temperature and humidity) have quite complex correlations with the strain/stress time variations, making it difficult to account for or eliminate them. Conversely, this work analyzed small time-scale structural events, i.e., the dynamic strain responses under short-term operating loads. The well-defined structural behavior thus permitted that the proposed approach could split the damage effects from the external load ones once the damage initiates, and at the same time, the low-rank property-based unsupervised machine learning tool should be a renewed valuable aid for splitting between the damage and masking effects. In addition, the data set in this study might be restricted by a series of single loadings in the experiment, so we are currently investigating the generalized frame from our two-step approach for more practical cases, such as strain responses obtained from the train-bridge interaction system.

## Figures and Tables

**Figure 1 sensors-21-06887-f001:**
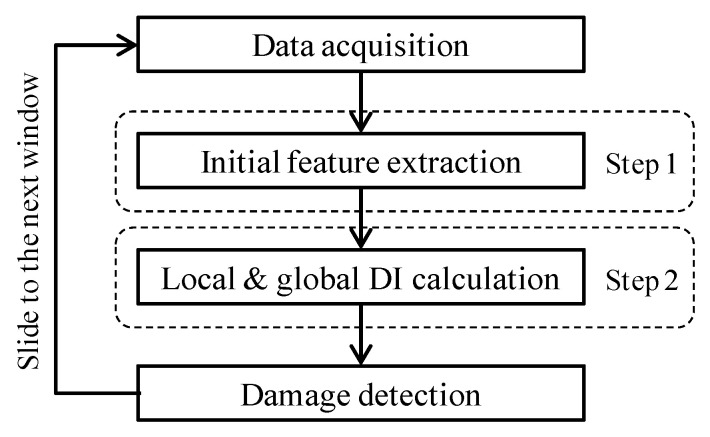
Schematic diagram of the output-only damage detection strategy covering the two-step approach to processing raw strain monitoring data.

**Figure 2 sensors-21-06887-f002:**
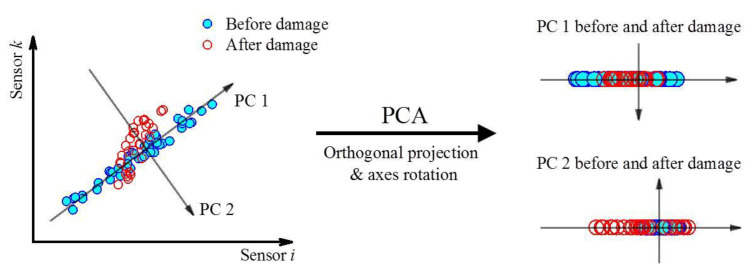
Graphical representation of a two-dimensional PCA. Circles represent strain observations in an undamaged state, and grey dots represent strain observations in a damaged state.

**Figure 3 sensors-21-06887-f003:**
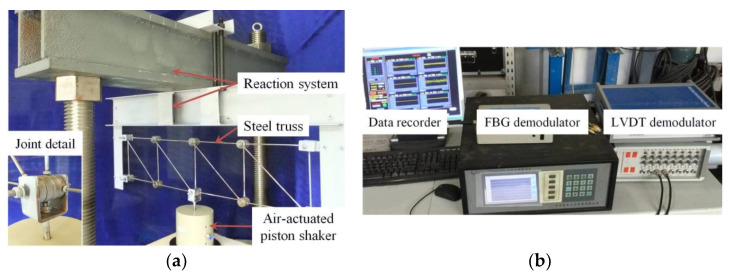
Experimental setup: (**a**) steel truss and loading device and (**b**) data acquisition system.

**Figure 4 sensors-21-06887-f004:**
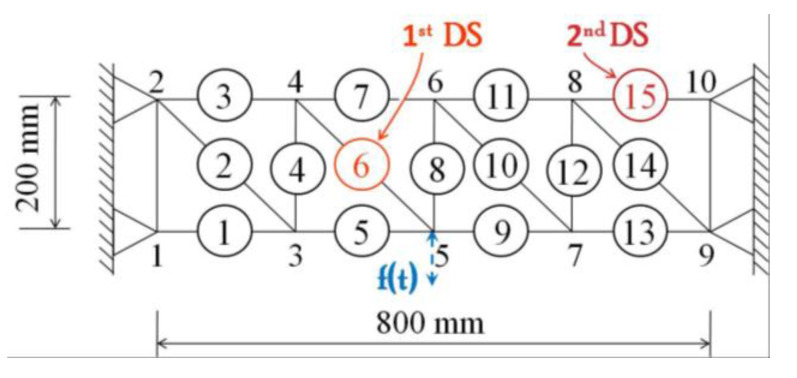
Experimental steel truss model with monitored member and damage position.

**Figure 5 sensors-21-06887-f005:**
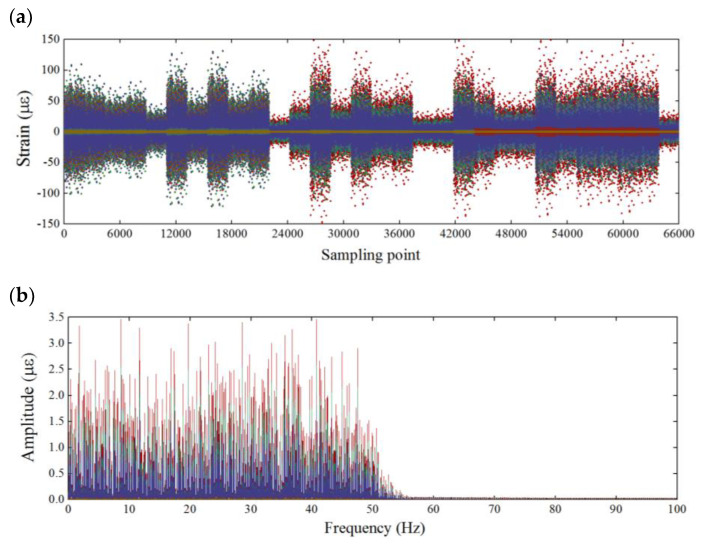
Raw strain monitoring data with the time-varying loads shown in (**a**) time domain and (**b**) frequency domain.

**Figure 6 sensors-21-06887-f006:**
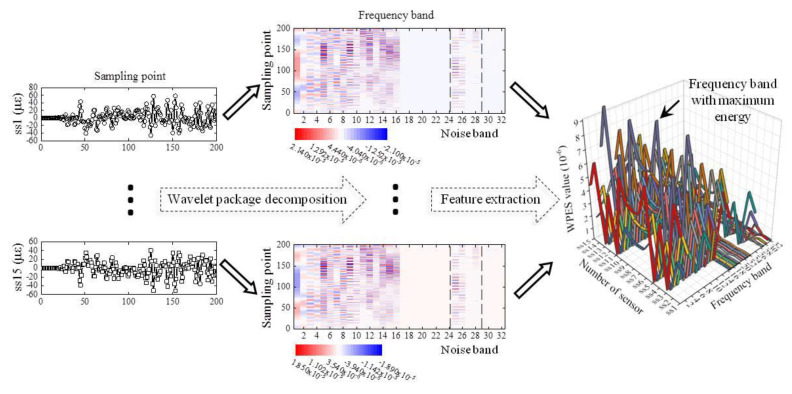
Example of WPES feature extraction of a data segment (200 sampling points from the first sliding time window of all the strain channels).

**Figure 7 sensors-21-06887-f007:**
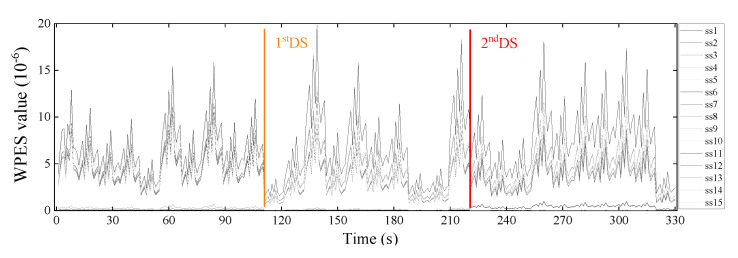
The extracted WPES values from raw strain monitoring data of all the strain sensors.

**Figure 8 sensors-21-06887-f008:**
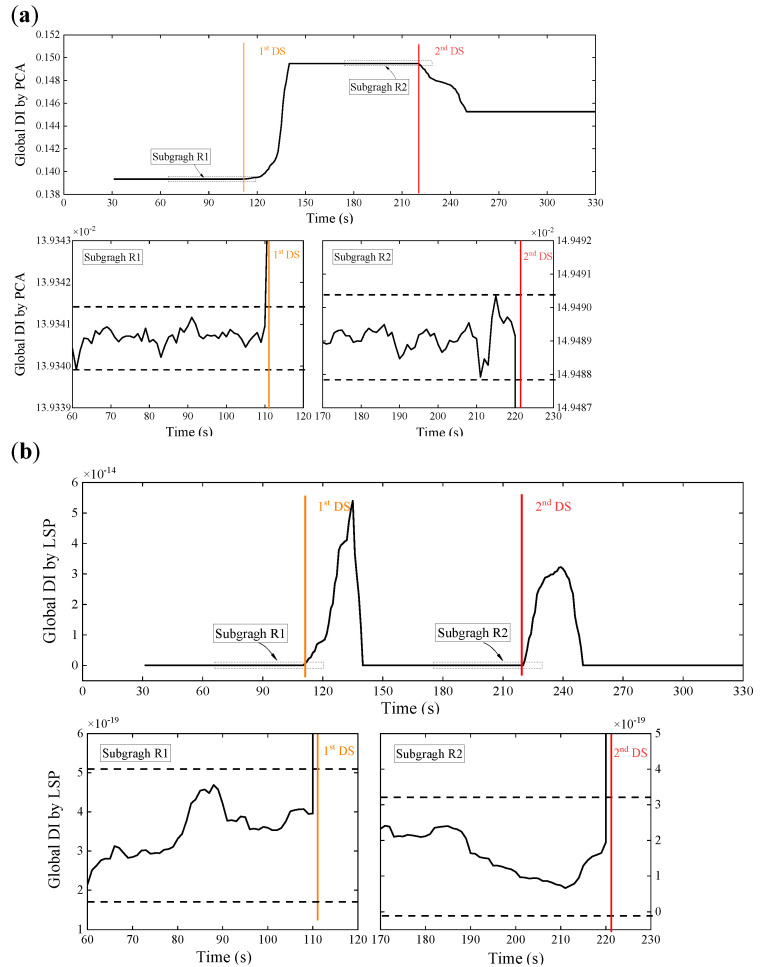
Results of damage detection using WPES-based (**a**) PCA and (**b**) LSP methods, respectively, with a window size of 30 s and a threshold of ±3 standard deviations over 80-s baseline data.

**Figure 9 sensors-21-06887-f009:**
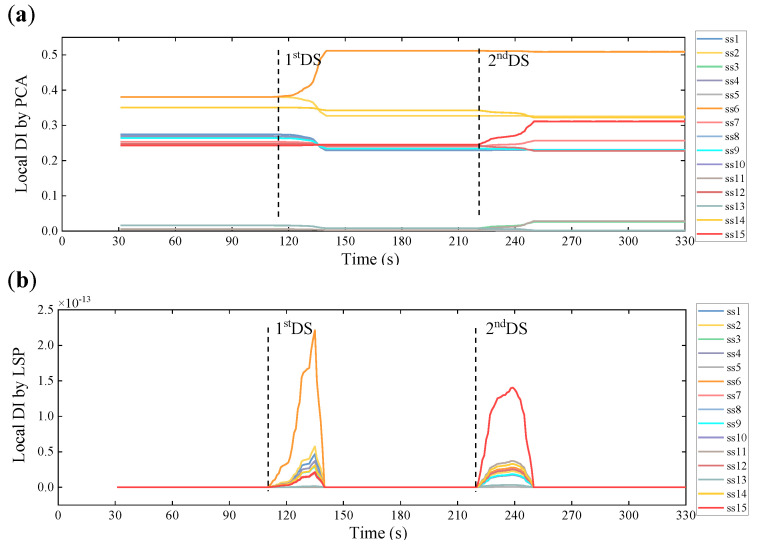
The temporal variation of local damage indicators based on the WPES-based (**a**) PCA and (**b**) LSP methods, respectively.

**Figure 10 sensors-21-06887-f010:**
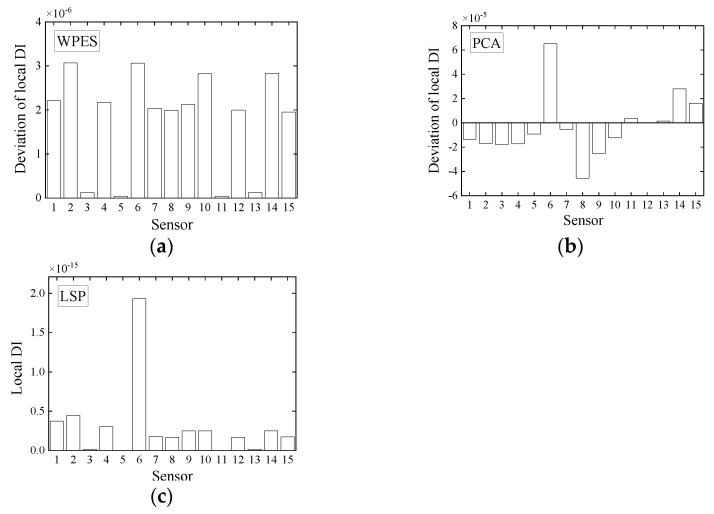
Deviations of local damage indicators corresponding to (**a**) WPES-only and WPES-based (**b**) PCA and (**c**) LSP methods respectively, when the first damage (1st DS) occurs (i.e., 111 s).

**Figure 11 sensors-21-06887-f011:**
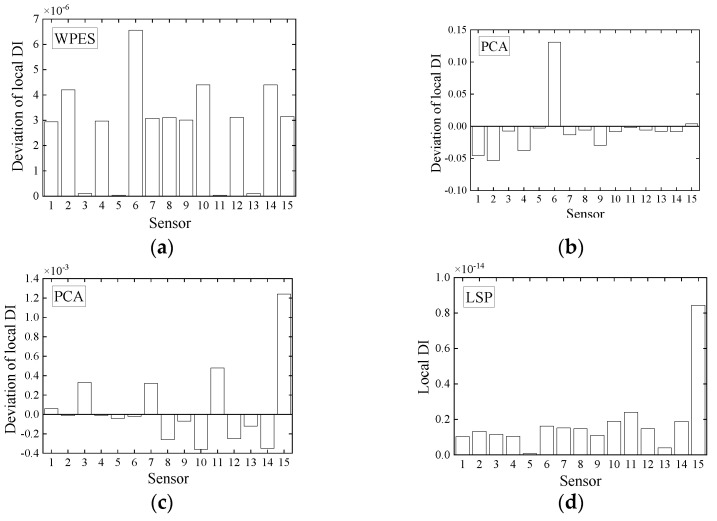
Deviations of local damage indicators corresponding to (**a**) WPES-only and WPES-based (**b**,**c**) PCA and (**d**) LSP methods respectively, when the second damage (2nd DS) occurs (i.e., 221 s). Note: WPES-based PCA method yields two results due to the different baselines used.

**Figure 12 sensors-21-06887-f012:**
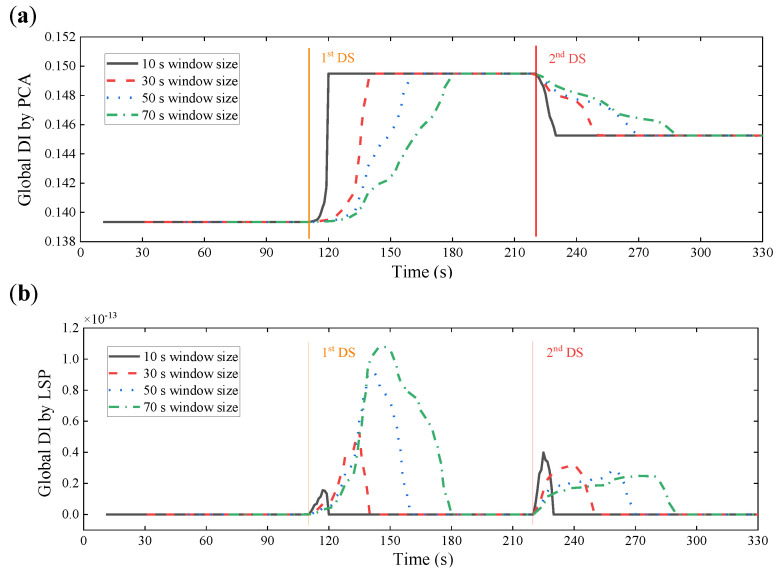
Results of damage detection using WPES-based (**a**) PCA and (**b**) LSP methods respectively, with different window sizes (10, 30, 50, and 70 s).

**Figure 13 sensors-21-06887-f013:**
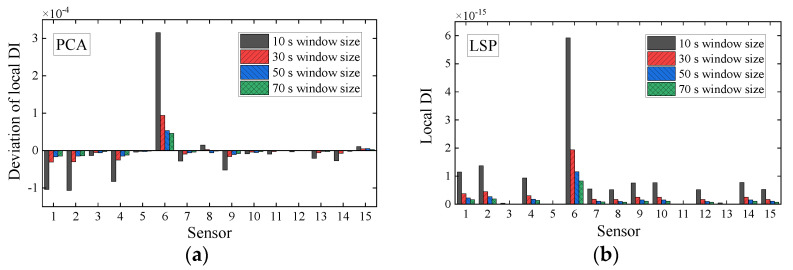
Deviations of local damage indicators corresponding to WPES-based (**a**) PCA and (**b**) LSP methods, respectively, when the first damage (1st DS) occurs (i.e., 111 s).

**Figure 14 sensors-21-06887-f014:**
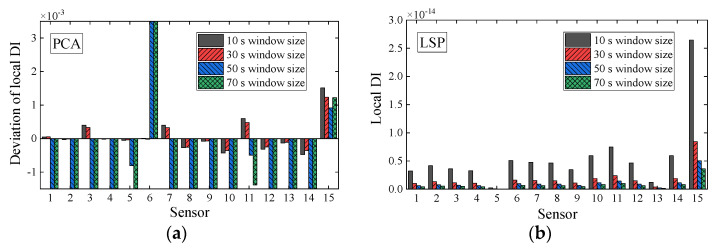
Deviations of local damage indicators corresponding to WPES-based (**a**) PCA and (**b**) LSP methods, respectively, when the second damage (2nd DS) occurs (i.e., 221 s).

**Figure 15 sensors-21-06887-f015:**
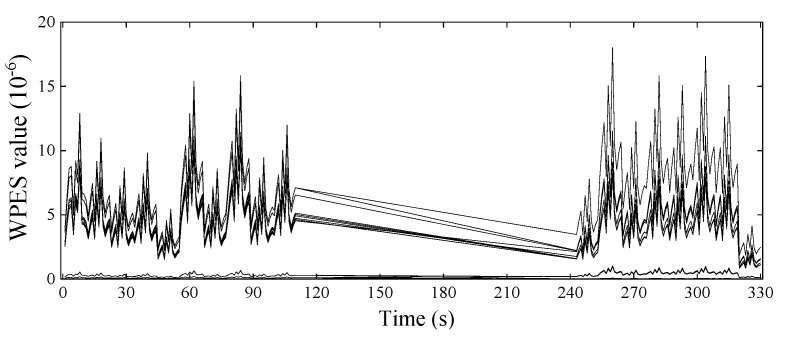
The temporal variation of WPES values containing a block of missing data.

**Figure 16 sensors-21-06887-f016:**
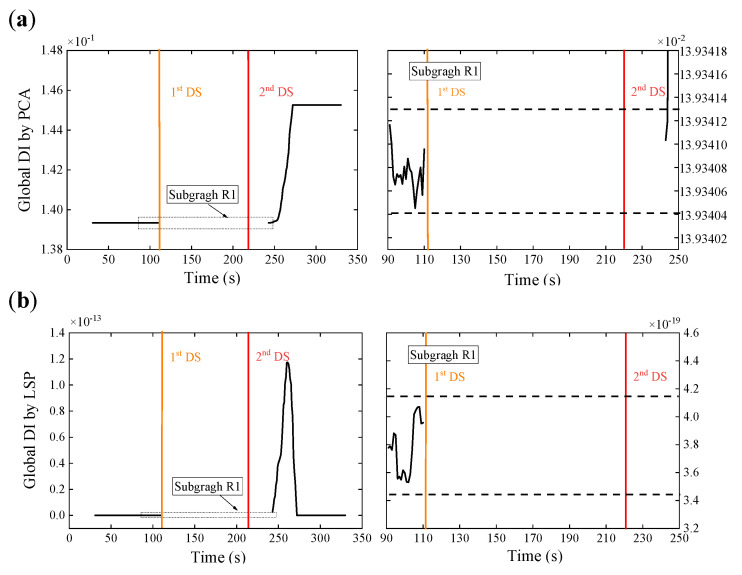
Results of damage detection for a data missing case referred to in Figure 7 and Figure 14. The WPES-based (**a**) PCA and (**b**) LSP methods are demonstrated, respectively.

**Figure 17 sensors-21-06887-f017:**
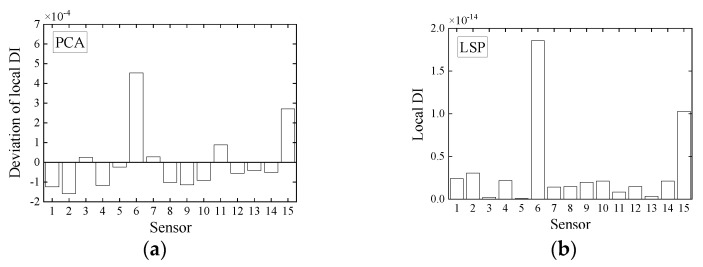
Results of local damage detection for a data missing case referred to in Figure 12. The WPES-based (**a**) PCA and (**b**) LSP methods are demonstrated, respectively.

**Figure 18 sensors-21-06887-f018:**
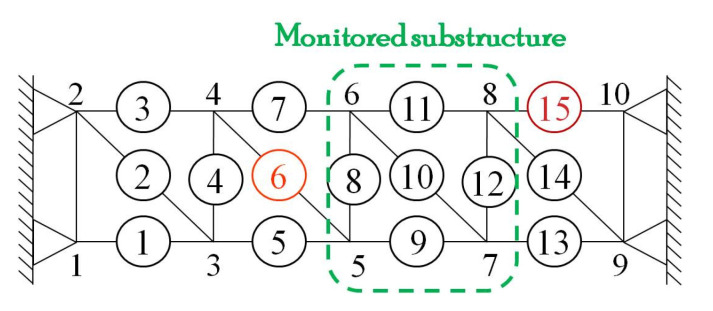
Experimental steel truss model with only a substructure monitored.

**Figure 19 sensors-21-06887-f019:**
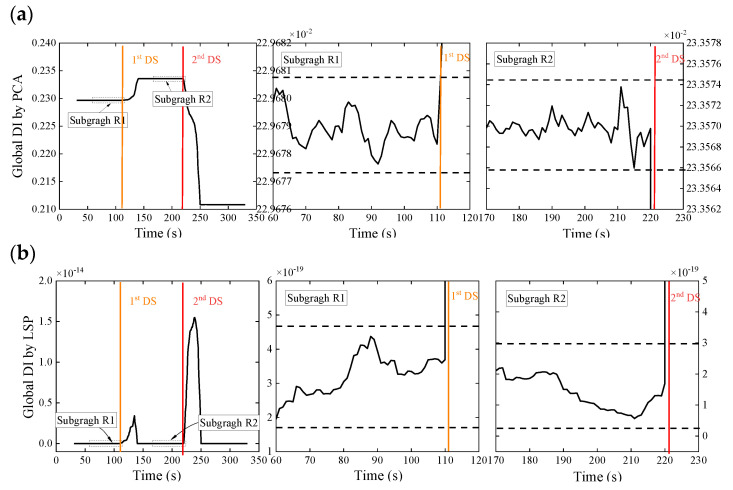
Results of damage detection for the case of the limited measuring points. The WPES-based (**a**) PCA and (**b**) LSP methods are demonstrated, respectively.

**Table 1 sensors-21-06887-t001:** Performance of damage detection using WPES-based PCA and LSP methods.

Damage Scenario	Time to Damage Initiation	Time to Detection		Clarity to Local Damage Identification ^1^	
		PCA	LSP	PCA	LSP
1st DS	111 s	111 s	111 s	Medium	High
2nd DS	221 s	221 s	221 s	Low	High

Superscript 1: three levels assessed for the capability of clarity to identify the damaged element. High level means identifying the damaged element clearly and directly with few positive faults; Medium level means identifying the damaged element indirectly or with negative faults; Low level means hard to identify or with many negative faults.

**Table 2 sensors-21-06887-t002:** Performance of damage detection with missing data (MD) and limited measurement (LM) cases using the WPES-based PCA and LSP methods.

Case	Time to Damage Initiation	Time to Detection		Clarity to Local Damage Identification	
		PCA	LSP	PCA	LSP
MD	243 s ^1^	245 s	243 s	Medium	High
LM (1st DS)	111 s	112 s	111 s	Low	Low
LM (2nd DS)	221 s	221 s	221 s	Low	Low

Superscript 1: the time when the measurement was resumed, at which time the damage already occurred.

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
