# Peer review of "Two-Step Approach to Processing Raw Strain Monitoring Data for Damage Detection of Structures under Operational Conditions"

_sensors, 2021, doi:10.3390/s21206887_

Round 1

Reviewer 1 Report

This paper presents a two-step approach to successively processing the raw strain monitoring data, consisting of the wavelet-based initial feature extraction step and the decoupling step to draw damage indicators. Principal component analysis and a low-rank property-based subspace projection method are adopted as two alternative decoupling methodologies.

The approach’s feasibility and robustness are substantiated by analyzing strain monitoring data from a customized truss experiment to successfully remove the masking effects of operating loads and identify local damages even concerning accommodating situations of missing data and limited measuring points.

In general, the paper is well-written. I have the following comments to improve the paper.

  1. It needs to be supplemented to explain whether the proposed method can achieve damage location and quantitative indication at the same time.
  2. What are the input variables and output variables in principal component analysis?
  3. The equation or expression of the new damage indicator (DI) should be given clearly in the paper.
  4. In actual engineering, which type of engineering structures can the proposed method be applied to?

Reviewer 2 Report

Notes in the attached file

Reviewer 3 Report

Very well written paper, I would like to see further discussion in the results - In my opinion it can be improved in this section. 
